# Rme1: Unveiling a Novel Repressor in the Cellulolytic Pathway of *Trichoderma reesei*

**DOI:** 10.3390/jof11090658

**Published:** 2025-09-06

**Authors:** Amanda Cristina Campos Antoniêto, David Batista Maués, Marcelo Candido, Iasmin Cartaxo Taveira, Roberto N. Silva

**Affiliations:** 1Department of Biochemistry and Immunology, Ribeirão Preto Medical School, University of São Paulo, Ribeirão Preto 14049-900, SP, Brazil; amanda.acc@gmail.com (A.C.C.A.); dbmaues@gmail.com (D.B.M.); marcelo.c@usp.br (M.C.); iasmin.cartaxo@gmail.com (I.C.T.); 2National Institute of Science and Technology in Human Pathogenic Fungi, São Paulo 14049-900, SP, Brazil

**Keywords:** *Trichoderma reesei*, Rme1, transcription factor, cellulase production, bioethanol

## Abstract

Thefilamentous fungus *Trichoderma reesei* is renowned for its exceptional ability to secrete cellulolytic enzymes, which play a crucial role in the hydrolysis of lignocellulose biomass. The expression of fungal cellulases is meticulously regulated at the transcriptional level, depending on the carbon source available in the medium. To obtain new insights into the transcriptional network controlling cellulase expression in *T. reesei*, we analyzed RNA-Seq data and identified and characterized a new transcription factor, Rme1, that regulates the expression of cellulolytic genes. Combining functional genomics and protein-DNA interaction assay, we showed that Rme1 acts as a repressor of cellulase production in *T. reesei* by directly regulating two critical genes involved in cellulose degradation: the cellobiohydrolase *cel7a* and the carbon catabolite repressor *cre1*. This is the first report of a transcription factor regulating Cre1. This study contributes to a better understanding of the complex regulation of the cellulolytic system of *T. reesei* and may be useful for the genetic modification of strains for the biorefinery industry.

## 1. Introduction

The filamentous fungus *Trichoderma reesei* is known for its high capacity to secrete cellulolytic enzymes that act in the hydrolysis of cellulose and is considered a model organism for cellulose degradation [1]. This species produces cellulolytic enzymes that participate in the deconstruction of the plant cell wall, which has been very interesting for the biorefinery industry, because the plant biomass constitutes an abundant renewable and conventional energy source for the food, paper and textile industries, as well as for the production of second-generation biofuels and biochemicals [2]. Therefore, *T. reesei* and other filamentous fungi are targets for studies aiming to understand and engineer their cellulolytic system to achieve better cellulase production.

The deconstruction of lignocellulosic biomass is intrinsically linked to the activity of holocellulolytic enzymes, including endoglucanases, exoglucanases or cellobiohydrolases, β-glucosidases, and xylanases and β-xylosidases, which work synergistically to degrade cellulose and hemicellulose polymers [3]. The cellobiohydrolase Cel7a is the most important cellulase produced by *T. reesei*, being the most abundant protein in the *T. reesei* secretome after growing in different lignocellulolytic substrates [4].

The gene expression and secretion of fungal cellulases are meticulously regulated at the transcriptional level. Small soluble sugars, such as lactose, cellobiose, sophorose, and xylose, play a crucial role in inducing cellulase expression in the model organism *T. reesei* [3]. Despite numerous studies on the regulation of cellulase expression in *T. reesei*, many transcription factors involved in this process still have unknown functions, or their interactions with DNA remain poorly understood. In recent years, there has been an increase in the characterization of new transcription factors that play a role in the cellulose degradation process [4,5,6], such as the positive regulators Xyr1, Ace3, Crz1, Azf1, Rxe1, Ace4 and Vib1, as well as the repressors Cre1, Rce1, and Rce2, among others [5].

Among the transcription factors identified in *T. reesei*, Xyr1 and Cre1 are the main regulators of cellulase and hemicellulase expression. Xyr1 is the primary transcriptional activator of these genes, playing a crucial role in inducing the production of hydrolytic enzymes in response to the presence of substrates such as cellulose and xylan [6,7]. This transcription factor recognizes promoter sequences of target genes and initiates their expression, facilitating the breakdown of complex polymers into simple sugars [8]. In contrast, Cre1 functions as the main repressor of cellulase and xylanase expression when glucose or other readily assimilable carbon sources are available [9]. This mechanism, known as carbon catabolite repression, occurs through Cre1 binding to *cis*-regulatory elements in the promoters of cellulase genes, thereby blocking transcription and ensuring that the fungus prioritizes more accessible energy sources [10]. The balance between Xyr1-mediated activation and Cre1-driven repression is essential for the metabolic adaptation of *T. reesei*, enabling it to regulate enzyme production according to environmental conditions and nutrient availability.

In silico analyses of RNA-seq data obtained by our group enabled the identification of conserved binding motifs in differentially expressed genes during the cultivation of *T. reesei* in cellulose, sophorose, and glucose, as well as *Aspergillus nidulans* grown in sugarcane bagasse (SCB) [11]. One conserved motif was associated with the *Saccharomyces cerevisiae* transcription factor Rme1p (Repressor of meiosis 1). In the yeast*,* Rme1p is a regulator of the cell cycle. It was first identified in a screening of mutations that affect mating or meiosis [12]. Further studies showed that Rme1p represses the expression of important regulators of meiosis, such as Ime1p and Ime2p [13]. Interestingly, the *T. reesei* homolog of Rme1, identified as the protein TRIREDRAFT_27649, was found to be four times more expressed during the cultivation of the *T. reesei* QM6a strain in sugarcane bagasse compared to glycerol [14], further supporting the potential role of this transcription factor as a regulator of holocellulolytic gene expression in *T. reesei*.

In the present work, we identified and characterized this new transcription factor, named Rme1 here. Using functional genomics, we investigated the role of Rme1 in *T. reesei* in the context of cellulase production. Our results showed that Rme1 acts as a repressor of cellulase gene expression. Protein-DNA interaction assay confirmed that Rme1 directly binds to the promoters of the *cel7a* and *cre1* genes. Therefore, Rme1 mediates the regulation of cellulolytic genes in both direct and indirect manners. Our work expanded the knowledge about the regulatory mechanisms governing cellulase production in *T. reesei*, increasing its biotechnological potential for biofuels and various other industrial applications.

## 2. Materials and Methods

### 2.1. Microbial Strains and Culture Conditions

*Escherichia coli* DH5α was used for plasmid propagation, while the Arctic strain was used for protein production. The strains were cultured on LB medium at 37 °C and 200 rpm. When necessary, 100 µg/mL ampicillin and 50 µg/mL gentamicin were added to the culture.

*T. reesei* strains were grown in MEX medium (3% [wt/vol] malt extract, 2% [wt/vol] agar) at 30 °C for 7–10 days until complete conidiation. For all experiments, 10^6^ conidia/mL for each strain were inoculated in Mandels-Andreotti medium [15] (MAM) containing the respective carbon source, and the cultures were incubated in an orbital shaker (200 rpm) at 30 °C for the indicated time. In the experiments using mycelia as inoculum, 10^6^ conidia/mL for each strain were inoculated in MAM containing 1% glycerol, and after 24 h, the mycelium was collected, washed with MAM without a carbon source, and transferred to MAM with 1% cellulose. The experiments were conducted in triplicate for each sample. After induction, the mycelia were collected by filtration, frozen, and stored at −80 °C. When necessary, 5 mM uridine was added to the cultures.

### 2.2. Identification of Rme1 and Phylogenetic and Structural Analyses

Analysis of RNA-Seq data from *T. reesei* and *A. nidulans* grown in different carbon sources for identification of putative *cis*-regulatory elements in the promoters of hydrolytic genes was performed as previously described [11]. Next, we used TOMTOM (https://meme-suite.org/meme/tools/tomtom, accessed on 18 May 2018) to search for similar known motifs from *S. cerevisiae* using the default parameters [16], which led to the Rme1p binding motif. BLASTp was used to identify the Rme1p homolog in *T. reesei*. Protein sequences of *T. reesei* and other fungi were obtained from FungiDB (https://fungidb.org/fungidb/app/, accessed on 5 August 2025). Amino acid sequence alignment was performed using ClustalW, and phylogenetic analysis was performed with MEGA X using the neighbor-joining method with 1000 bootstraps [17]. For structural analysis, protein sequences were submitted to SMART (http://smart.embl-heidelberg.de/, accessed on 15 August 2019) [18].

### 2.3. Construction of Deletion Cassette and Fungal Transformation

Homologous recombination in yeast was used to construct the *rme1* deletion cassette. Briefly, a 1 kb fragment of the 5′-UTR and 3′-UTRs was amplified from *T. reesei* genomic DNA (gDNA) using the primers Rme 5′F/R and 3′F/R, respectively, along with the selection marker *pyr4* (TRIREDRAFT_74020), amplified with the primers Rme_Pyr4 F/R. The 5′-UTR, 3′-UTR, and *pyr4* fragments were purified and used to transform the *S. cerevisiae* SC9721 strain, along with the plasmid pRS426, linearized with EcoRI and XhoI, using the lithium acetate method [19]. The *rme1* deletion cassette was PCR-amplified from gDNA extracted from the respective *S. cerevisiae* transformant using the primers Rme 5′ F and Rme 3′ R. All the PCRs were performed using Phusion High Fidelity Polymerase, and all the primers are listed in Appendix A.

To obtain the *rme1* knockout strain, protoplasts from the strain QM6aΔ*tmus53*Δ*pyr4* were obtained by mycelium digestion with Lysing enzymes from *Trichoderma harzianum* (Sigma, L1412, Darmstadt, Germany ) and mixed with approximately 40 µg of linear deletion cassette. Polyethylene glycol (PEG)-mediated protoplast transformation was carried out as previously described [20]. Transformants were plated on MAM containing 2% glucose, and 1.2 M Sorbitol, without uridine, and incubated at 30 °C until colonies were observed. The candidates were submitted to three rounds of selection on MAM with and without 0.1% Triton X-100 (GE Healthcare, Uppsala, Sweden). Deletion of *rme1* was confirmed by diagnostic PCR and RT-qPCR (Appendix A) with primers listed in Appendix A, respectively.

### 2.4. Evaluation of the Growth and Conidiation Profile of the Δrme1 Strain

To evaluate the growth profile of Δ*rme1* in different carbon sources, the WT and mutant strains were grown on MAM plates containing 1% (*w*/*v*) of carbon source (glucose, xylose, mannose, starch, and glycerol) at 30 °C. The growth of the WT and Δ*rme1* strains was recorded on the 4th and 7th days. Analysis of the conidiation profile was performed in race tubes with MEX medium over 10 days. All experiments were performed in triplicate.

### 2.5. Gene Expression Analysis by RT-qPCR

Mycelia from the WT and Δ*rme1* strains were macerated, and the RNA was extracted using TRI^®^ (Sigma-Aldrich, St. Louis, MO, USA), according to the manufacturer’s instructions. For cDNA synthesis, 1 µg of RNA was first treated with DNAse I (ThermoFisher Scientific, Waltham, MA, USA) to remove genomic DNA. After this step, cDNAs were synthesized using Maxima^TM^ First Strand cDNA Synthesis (ThermoFisher Scientific, Waltham, MA, USA) according to the manufacturer’s instructions. They were diluted 50× and used for RT-qPCR analysis in Bio-Rad CFX96^TM^ equipment with SsoFast EvaGreen Supermix (Bio-Rad, Hercules, CA, USA), according to the manufacturer’s instructions. Gene expression levels were calculated from the threshold cycle according to the 2^−ΔCT^ method relative to transcript levels of β-Actin [21,22]. The amplification program used in this study was as follows: 95 °C for 10 min followed by 39 cycles of 95 °C for 10 s and 60 °C for 30 s followed by a dissociation curve of 60 °C to 95 °C with an increment of 0.5 °C for 10 s per increment. The primers used for amplification of identified genes are described in Appendix A. Statistical tests were performed using one-way ANOVA (and nonparametric testing), followed by Bonferroni’s test (to compare all pairs of columns) (available in Prism software v. 6.0) for comparing the gene expression levels of the WT and mutant strains.

### 2.6. Protein and Enzyme Assays

Mycelia from the WT and Δ*rme1* strains grown in cellulose for 48 h (after pregrowing in glycerol for 24 h) were used for protein extraction. Mycelia were grounded using liquid N_2_ and added to 800 µL of extraction buffer [137 mM NaCl, 2.7 mM KCl, 10 mM Na_2_HPO_4_, 1.8 mM KH_2_PO_4_, 1× Protease Inhibitor Mix (GE Healthcare)]. Samples were sonicated (60% amplitude, pulse 10 s on/10 s off, 1 min) and centrifuged at 13,000 rpm for 20 min at 4 °C. The supernatant was collected for enzyme assays. β-glucosidase activity was determined by the addition of 50 µL of 50 mM sodium acetate buffer (pH 5.5), 10 µL of the sample, and 40 µL of 5 mM p-nitrophenyl (PNP)-glucoside substrate. The reaction mixture was incubated at 50 °C for 15 min, followed by adding 100 µL of 1 M sodium carbonate. To analyze cellobiohydrolase activity, 50 µL of 50 mM sodium citrate (pH 4.8) was incubated with 10 µL of the sample and 40 µL of 5 mM PNP-cellobioside. The reaction mixture was incubated at 50 °C for 3 h, followed by adding 100 µL of 1 M sodium carbonate. Samples were read at an absorbance of 405 nm. One enzyme unit was defined as the amount of enzyme capable of releasing 1 µmol of reducing sugar or hydrolyzing 1 µmol of substrate per minute. Protein concentration was determined using Qubit Protein Assay (Thermo Fisher Scientific). Statistical tests were performed using Student’s *t*-test (and nonparametric testing), followed by Welch’s test (available in Prism software v. 6.0) for comparing the gene expression levels of the WT and mutant strains. Salts were purchased from Synth (Synth, Diadema, SP, Brazil) and PNP substrates were purchased from Sigma (Sigma-Aldrich, St. Louis, MO, USA).

### 2.7. Prediction of Rme1 Binding Motif and Search for Its Putative Targets

The putative Rme1 binding motif was obtained from the online server http://zf.princeton.edu/index.php (accessed on 20 January 2021) [23]. Promoter sequences comprising 1 kb upstream of the ATG were obtained by an ad hoc script. A search for putative Rme1 binding sites in the promoter of the target genes was performed using FIMO [24], available on MEME [16], to identify possible Rme1 direct targets.

### 2.8. Docking Analysis

Modeling of protein and DNA structures and molecular docking were performed in the HDOCK webserver (http://hdock.phys.hust.edu.cn/, accessed on 10 July 2024) [25], using sequences retrieved from JGI (the Rme1 full sequence was retrieved from the *T. reesei* QM6a genome version). To model the promoter sequences, 15 nucleotides before and 15 nucleotides after the Rme1 putative binding motif were included to achieve a better simulation. Since HDOCK generated a model with only part of the protein docked to the nucleotide sequence, the Robetta WebServer (https://robetta.bakerlab.org/, accessed on 10 July 2024) was used to predict the full Rme1 structure. The model was further validated in the MolProbity webserver (http://molprobity.biochem.duke.edu, accessed on 10 July 2024).

### 2.9. Heterologous Expression of Rme1 in E. coli

The cDNA of *T. reesei* grown on sugarcane bagasse was used as a template to amplify the Rme1 ORF using the primers RmeF_BamHI and RmeR_XhoI (Appendix A). The fragment was purified and digested with BamHI and XhoI and ligated into the vector pGEX-4T-1, previously digested with the same enzymes, using the T4 Ligase (Thermo Fisher Scientific, Waltham, MA, USA), generating the plasmid pGEX-Rme1. Absence of mutations was confirmed by Sanger sequencing.

The *E. coli* Arctic (DE3) was used as a host for protein production. The expression of Rme1:GST was induced with 100 µM IPTG, and the culture was maintained at 16 °C for 24 h for protein production. After that, cells were lysed, and the Rme1:GST protein was purified using GST Sepharose 4B GST-Tagged protein purification resin following the manufacturer’s instructions (Appendix A).

### 2.10. Electrophoretic Mobility Shift Assay (EMSA)

For EMSA, probes covering 300 bp of the P*cel7a* and P*cre1* promoters were amplified from *T. reesei* gDNA using the primers Cel7a F/R and Cre1 F/R (Appendix A), respectively, and purified using Amicon Ultra Centrifugal Concentration (Millipore, Darmstadt, Germany). For the negative control, a probe for the *swo* gene covering 140 bp of its promoter was amplified using the primers Swo F/R (Appendix A). Binding assays were carried out using different concentrations of Rme1 and 100 ng of the probe in the Interaction Buffer (20 mM Tris-HCl pH 7.4, 50 mM KCl, 5% glycerol, 1 mM DTT, 50 µg/mL BSA, 50 µM Zinc Sulfate) for 25 min at room temperature. After incubation, protein-bound and free DNAs were separated by electrophoresis in nondenaturing 6% polyacrylamide gels with 0.5% Tris-Borate running buffer at 4 °C. Gels were stained with SYBR™ Gold Nucleic Acid Gel Stain (Thermo Fisher Scientific, Waltham, MA, USA) for 25 min and then visualized at ChemiDoc equipment (Bio-Rad, Hercules, CA, USA).

## 3. Results

### 3.1. Identification of TRIREDRAFT_27649 as a Putative Regulator of Cellulase Production in T. reesei

We had previously performed an RNA-Seq analysis and identified conserved binding motifs in the promoters of hydrolase-encoding genes that are differentially expressed under several growth conditions. One enriched motif was associated with the *S. cerevisiae* transcription factor Rme1p (Figure 1a). Although the found motif does not present a high degree of conservation with the Rme1p binding motif, it is possible to observe that both present the 5′-TCAAAA-3′ consensus sequence. Next, we used the Rme1p protein sequence to perform a BLASTp search in the *T. reesei* genome, indicating the gene TRIREDRAFT_27649 as the putative homolog of the Rme1p (identity 28%, e-value 6e-06, mostly in the DNA binding domain). Hereafter, we named the gene TRIREDRAFT_27649 as *rme1*. A phylogenetic analysis of the Rme1 homologs in various fungi species indicates that this protein has diverged considerably in the course of evolution (Figure 1b). Structure analysis by SMART showed that the first change is the size of the proteins: while the proteins found in filamentous fungi have around 600 amino acids, Rme1p has half the size, and the homolog from *Candida albicans* is even smaller, having only 250 aa (Figure 1c). Another remarkable change is the number of C_2_H_2_ zinc finger domains: the TFs from filamentous fungi have four zinc fingers, while Rme1p has three, and the *C. albicans* homolog has only two (Figure 1c). Interestingly, the arrangement of these zinc fingers seems conserved between Rme1p and its orthologs from filamentous fungi, where the last one is slightly distant from the others. Taken together, these results suggest that a divergence in these proteins has occurred during evolution, which can lead to the development of novel regulatory functions of Rme1p homologs in filamentous fungi.

Next, we accessed the gene expression of Rme1 in RNA-Seq data from our research group. The *rme1* gene presented a log_2_ fold change (log_2_FC) of 1.9 (adjusted *p*-value < 0.05) when *T. reesei* was grown in the inducer carbon source sugarcane bagasse when compared to glycerol [14]. We used RT-qPCR to profile the *rme1* gene expression across different carbon sources and different time points. As shown in Figure 1d, *rme1* expression is not carbon source-dependent, being very constant under these conditions. There is no statistical difference in the expression levels between the majority of the tested conditions; however, *rme1* expression significantly increases at 24 h of growth on cellulose, compared to the expression at 8 h on the same carbon source (*p*-value < 0.05, as accessed by One-way ANOVA followed by Bonferroni’s test).

### 3.2. Rme1 Is Required for Normal Conidiation in T. reesei

A knockout strain was constructed to further study the function of *rme1* in *T. reesei* (Appendix A). The first observation was that this TF is important for conidiation in the fungus. The Δ*rme1* strain presents more aerial conidia and also shows some defects in the production of the typical green pigment present in *T. reesei* conidia (Figure 2a). This is more evident after a long period of growth (Figure 2a, bottom panel). Race tube assay showed that the deletion of *rme1* also causes a delay in conidiation (Figure 2b). The WT and Δ*rme1* strains were also grown in different carbon sources (starch, glycerol, mannose, xylose, glucose); however, no significant differences were observed between the two strains during the experiment, except for the growth on xylose, where the Δ*rme1* strain grew better than the WT (Figure 2c). These results indicate that Rme1 is important for conidiation in *T. reesei* but is dispensable for growth in several carbon sources.

### 3.3. Rme1 Controls the Expression of Holocellulases in T. reesei

To assess whether Rme1 has a role in regulating the expression of hydrolytic genes, we first performed RT-qPCR to analyze the expression of the main cellulolytic genes in the WT and Δ*rme1* strains. The strains were grown in cellulose to induce the cellulase expression, and also in glucose, to evaluate whether Rme1 has a function in repressing conditions. Our results showed that the levels of expression of the cellobiohydrolases *cel7a* and *cel6a*, as well as the β-glucosidase *cel1a*, and the sugar transporter *Tr69957*, are significantly increased in the Δ*rme1* strain (Figure 3a–d). Interestingly, this increase is only observed when the strains were grown in cellulose, while in glucose, the expression of these genes is undetectable for both strains.

In *T. reesei*, the main repressor of cellulase expression is the carbon catabolite repressor Cre1. To investigate whether there is a relationship between Rme1 and Cre1, we included the latter in our gene expression analysis. As shown in Figure 3e, deletion of *rme1* negatively impacts the expression of *cre1* in both cellulose and glucose cultivations. Therefore, Rme1 is important for the appropriate expression of *cre1*.

The analysis of the expression of ten holocellulases demonstrated that most of them are up-regulated in the Δ*rme1* strain in the presence of cellulose (Figure 4a). We included the neutral carbon source glycerol in this analysis to see the behavior of Rme1 in different carbon sources, further confirming that repression of holocellulase expression mediated by Rme1 only occurs in inducing conditions (cellulose). Expression of *cre1* is decreased in the Δ*rme1* strain in all the tested conditions, while expression of *xyr1* is unaffected (Figure 4a). We also measured the cellulase activity to verify if it is affected by Rme1. As shown in Figure 4b,c, both β-glucosidase (Figure 4b) and cellobiohydrolase (Figure 4c) are significantly higher in the Δ*rme1* strain. Taken together, these results demonstrate that Rme1 is a repressor of the expression of holocellulolytic genes, and this repression occurs during cellulose breakdown and is partially mediated by Cre1, causing an overall induction of the main cellulases produced by *T. reesei*.

### 3.4. Rme1 Binds to the Promoters of cel7a and cre1 Genes

Our results showed that the deletion of *rme1* affects the expression of several holocellulolytic genes. As Rme1 encodes a C_2_H_2_ transcription factor, it was possible to predict its binding motif through bioinformatic analysis (Figure 5a). The Rme1 binding motif seems conserved with its ortholog Rme1p from *S. cerevisiae* (Figure 1a and Figure 5a), and both present the core sequence 5′-GNGG-3′. Using MEME, we searched for Rme1 putative binding sites in the promoters of the genes evaluated in the gene expression analysis. Only two genes presented putative binding sites for Rme1 in their promoters (*p*-value < 0.0001): *cel7a* and *cre1* (Figure 5b). Indeed, deletion of *rme1* increases the expression of *cel7a* (Figure 3a and Figure 4) while decreasing the expression of the transcription factor *cre1* (Figure 3e and Figure 4), reinforcing that these genes are important targets of Rme1.

Next, we conducted molecular docking simulations to analyze the possible interaction between Rme1 and the P*cel7a* and P*cre1* promoters. The docking results show that Rme1 can recognize and bind to its predicted binding motif in both promoters (Figure 5c,d), presenting a good Docking score and a high Confidence score (Confidence score above 0.7 means that the two molecules would be very likely to bind) for both cases. The docking analyses also confirmed that these interactions involve the Rme1 DNA binding domain and its binding sites in the promoters (Appendix A). These results strongly suggest that Rme1 can directly bind to the promoters of *cel7a* and *cre1* genes.

In order to confirm whether Rme1 can directly regulate these genes, we carried out an Electrophoretic Mobility Shift Assay (EMSA) to analyze the interaction between Rme1 and the promoters of these two genes. Probes covering the promoter regions containing the Rme1 putative binding sites for both the P*cel7a* and *Pcre1* promoters were used in the assay. In the EMSA, the recombinant Rme1 protein binds to the promoters of both the *cel7a* and *cre1* genes in a typical protein concentration-dependent manner, where the retardation occurs upon addition of 1.6 µM of Rme1 (Figure 6a,b). As a negative control, the EMSA was performed using a probe for the *swo* gene and increasing concentrations of purified Rme1; however, no shift in mobility was observed (Appendix A), indicating no formation of a protein-DNA complex and that the interaction between Rme1 and P*cel7a* and P*cre1* was validated. However, additional studies employing ChIP-seq and surface plasmon resonance (SPR) are required to determine the binding affinity of Rme1 to *Pcel7a* and *Pcre1*. These results showed that Rme1 regulates the expression of the *cel7a* and *cre1* genes by directly binding to their promoters.

## 4. Discussion

The discovery of new transcription factors in *T. reesei* is crucial for advancing biotechnological applications. *T. reesei* is a filamentous fungus renowned for its ability to produce a wide array of hydrolytic enzymes that break down lignocellulosic biomass into simple sugars, which can then be fermented into value-added products, such as fine chemicals and bioethanol [26]. Due to their important function in regulating biomass deconstruction, transcription factors are important targets for genetic engineering of new strains with enhanced cellulase production, through different approaches: protein engineering, deletion or overexpression, promoter engineering, among others [1]. Here, we combined RNA-Seq analysis, comparative genomics, functional genomics, and protein-DNA interaction assays to identify and elucidate the role of a new transcription factor in the *T. reesei* cellulolytic pathway. Using this approach, we had previously identified another TF, Azf1, and characterized its role in cellulase production by *T. reesei* [11]. Therefore, the search for putative *cis*-regulatory elements in the promoter of genes of interest is a promising approach to identify new regulators.

Our RNA-Seq analysis led us to a putative *cis*-regulatory element enriched in the promoters of holocellulolytic genes differentially expressed in several growth conditions. This motif was associated with the *S. cerevisiae* transcription factor Rme1p. Using BLASTp, we identified a putative Rme1p homolog in *T. reesei*. However, our phylogenetic and structural analyses showed that the Rme1p orthologs from filamentous fungi have diverged considerably from those of yeast during evolution (Figure 1b,c). This divergence could increase novel functions associated with this regulator. Indeed, our work shows that, in *T. reesei*, Rme1 is involved in regulating holocellulase expression, an unreported role for this TF in fungi. While in *S. cerevisiae*, Rme1p is involved in cell cycle regulation [12], the role of its orthologs in filamentous fungi has been repurposed. In the pathogenic fungus *Aspergillus fumigatus*, ZfpA, the Rme1 homolog, is involved in hyphal branching, stress response, virulence, and antifungal response [27]. In *Aspergillus flavus*, ZfpA is involved in asexual and sexual development, secondary metabolite production, and overexpression of *zfpA* positively impacts conidiation in this fungus [28]. It seems that the role of Rme1 and its orthologs in conidiation is conserved in some filamentous fungi. These results showed that the role of Rme1p orthologs in filamentous fungi had expanded to new biological functions. In *T. reesei*, several factors affect conidiation, such as light, pH, calcium, and carbon, nitrogen, and oxygen availability [29]. Some transcription factors have a role in regulating both conidiation and holocellulase expression in *T. reesei*, such as Vel1 [30] and Azf1 [11], besides Rme1 itself. Although the relationship between these two physiological processes needs to be deeply investigated, these results suggest that these processes can be controlled by the same regulators, probably as a response to nutrient status in the environment.

Besides regulating conidiation in *T. reesei*, our results also demonstrated that Rme1 regulates the expression of hydrolytic genes. Deletion of this TF increases the expression of several genes associated with biomass deconstruction: the cellobiohydrolases *cel7a* and *cel6a*, the endoglucanase *cel5a*, the β-glucosidases *cel1a* and *cel3b*, and the xylanase *xyn2* (Figure 3 and Figure 4a), showing that Rme1 acts as a repressor of holocellulase expression. Expression of the sugar transporter *Tr69957* is also higher in the Δ*rme1* strain (Figure 3d). *Tr69957* can transport xylose, cellobiose, and mannose [31], which can explain why the Δ*rme1* strain grows better in xylose as a carbon source than the WT (Figure 2c). Furthermore, our results showed that the function of Rme1 in regulating holocellulase expression is carbon source-dependent: this TF only represses the expression of these genes during biomass breakdown. In addition, the results indicate that the absence of Rme1 is not sufficient to activate the expression of cellulases without an inducer, as we did not observe an increase in the expression of cellulolytic genes in the Δ*rme1* strain when grown in glycerol or glucose. Our expression analyses also revealed that expression of *cre1* is partially dependent on Rme1 (Figure 3e and Figure 4), suggesting that these two transcription factors act together to repress the expression of hydrolytic genes. Enzymatic activities showed that Rme1 affects cellulase production not only at the transcriptional level but also at the protein level, as its deletion increases cellobiohydrolase and β-glucosidase activities (Figure 4b,c).

To obtain more insights into the mechanisms used by Rme1 to regulate holocellulase expression, we performed in silico analyses to identify possible targets of this TF. Motif search and docking analyses strongly suggested that Rme1 binds to the promoters of *cel7a* and *cre1* genes (Figure 5b–d). To validate these results, we performed EMSA to analyze the interaction between the recombinant Rme1 and these promoters. Our results showed that Rme1 can bind to these promoters in vitro (Figure 6). DNA-protein complexes were observed for both promoters; however, they needed a high amount of Rme1 to reduce their mobility, suggesting that Rme1 possibly has a low affinity for these promoters in vitro. Other experiments, such as Surface Plasmon Resonance and Chromatin Immunoprecipitation, can analyze the affinity of Rme1 to these promoters and their interaction in vivo, respectively. Despite this, the EMSA combined with the gene expression and docking analyses, reinforces the notion that Rme1 can bind to the promoters of these genes to directly regulate their expression. It is noteworthy that Rme1 can function as an activator or a repressor, depending on its target gene. This is probably due to specific signaling pathways and protein–protein interactions. The transcription factor Cre1/CreA/Mig1p is known for being a transcriptional repressor. In *Aspergillus nidulans*, an integrated RNA-Seq and ChIP-Seq analysis identified CreA target genes that are positively regulated by it [32]; however, the mechanism by which CreA can activate their expression still needs to be investigated. Interestingly, the Cyc8/Tup1 transcriptional co-repressors, which are partners of CreA/Mig1p in the repression of its target genes, also interact with Cre1 in *T. reesei* and with Xyr1, the main activator of cellulase gene expression [33,34,35]. Furthermore, Cyc8/Tup1 are required for appropriate cellulase expression, functioning also as co-activators for Xyr1 [34]. These studies indicate that transcription factors and other DNA-binding proteins can have different regulatory functions depending on cellular context.

As Cel7a is the most important cellulase produced by *T. reesei*, its expression is highly regulated. The *cel7a* gene is a target of several transcription factors, such as the activators Xyr1, Crz1, Ace3, Ace4, Azf1, among others [11,36,37,38], and the repressors Rce1 and Rce2 [39,40]. Interestingly, Rce1 and Rce2 compete with Xyr1 and Ace3, respectively, for binding to the *cel7a* promoter [39,40]. Our work revealed Rme1 as a novel repressor of *cel7a* expression, and its deletion dramatically increases cellobiohydrolase activity in *T. reesei* (Figure 4c). Cre1 is an important regulator of holocellulase expression in *T. reesei*, being the main repressor of their expression. However, the regulation of Cre1 is not well studied in this fungus. Our results show that expression of *cre1* requires Rme1, and to the best of our knowledge, this is the first report of a transcription factor directly regulating *cre1* expression in *T. reesei*.

The integration of RNA-Seq analysis, functional genomics, and protein-DNA interaction assay allowed us to identify a novel regulator of cellulase expression in *T. reesei* and partially characterize its regulatory mechanisms. Our data demonstrate that Rme1 regulates cellulose degradation by *T. reesei* through directly regulating the expression of critical genes in the *T. reesei* cellulolytic pathway: *cel7a* and *cre1*. To summarize the role of Rme1 in regulating cellulase expression, we propose a model where the glucose, released during the cellulose breakdown by the action of cellulases, activates a signaling pathway that, in turn, leads Rme1 to the promoter of *cre1* to activate its expression, and together, these two repressors bind to the promoters of holocellulases genes, such as *cel7a*, to repress their expression (Figure 7). Our work highlighted Rme1 as a novel target for *T. reesei* engineering to enhance cellulase production. Deletion of this TF elevates cellulase production, with a highlight for β-glucosidases, the least produced enzyme by *T. reesei*. The protein-DNA results can be used to drive strategies such as promoter engineering and transcription factor engineering by domain fusion. For example, replacement of the Rme1 binding site in the *cel7a* promoter or fusion of the Rme1 DNA-binding domain to a trans-activator domain can increase *cel7a* expression. This can lead to strains with higher cellulase production.

## 5. Conclusions

Using different approaches, we identified and characterized a new regulator of cellulase expression in *T. reesei*. Rme1 acts as a repressor of cellulase expression by directly repressing the expression of the main cellulase produced by *T. reesei*—Cel7a—and activating the expression of the main repressor Cre1. Our findings identified Rme1 as a new target for genetic and promoter engineering in *T. reesei*, aiming to generate strains with higher production of cellulolytic enzymes.

## Figures and Tables

**Figure 1 jof-11-00658-f001:**
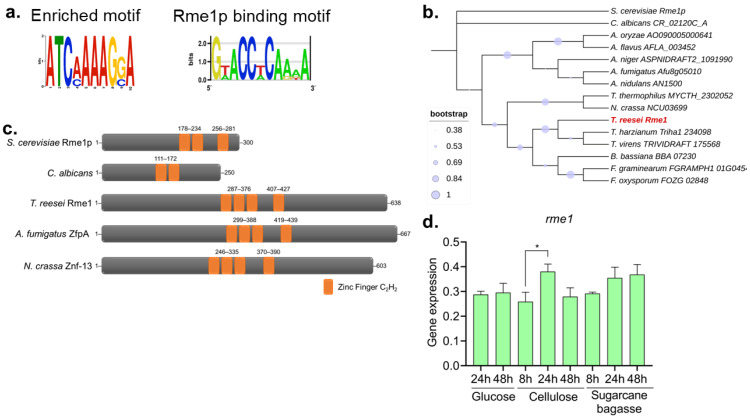
Identification of Rme1 in *T. reesei*. (**a**) Enriched motif found in the promoters of cellulase-encoding genes differentially expressed in RNA-Seq from *T. reesei* grown in different carbon sources (left) and the Rme1p binding motif (right). (**b**) Cladogram of Rme1 homologs in fungi. Sequences were obtained from FungiDB, and the analyses were performed using MEGA. *T. reesei* Rme1 is highlighted in red. (**c**) SMART structural analysis of Rme1 homologs from *S. cerevisiae*, *C. albicans*, *T. reesei*, *A. fumigatus*, and *N. crassa*. The C_2_H_2_ zinc finger is highlighted in orange. (**d**) Gene expression profile of *rme1* accessed by RT-qPCR. *T. reesei* was grown on MAM containing glucose, cellulose, or sugarcane bagasse for the indicated time. For growth on glucose, conidia (10^6^/mL) were inoculated in MAM. For cellulose and sugarcane bagasse growth, conidia (10^6^/mL) were inoculated in MAM containing glycerol and cultivated for 24 h before the formed mycelia were transferred to new media containing the respective carbon source (* *p*-value < 0.05).

**Figure 2 jof-11-00658-f002:**
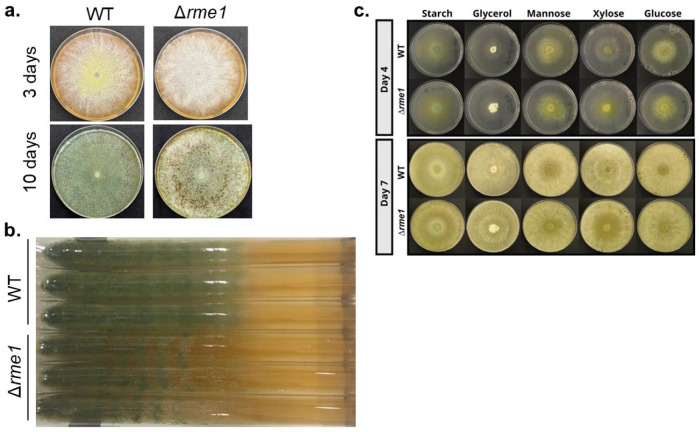
Rme1 is important for conidiation in *T. reesei*. (**a**) Conidia (10^6^/mL) from WT and Δ*rme1* strains were inoculated in MEX plates and incubated at 30 °C for the indicated time. (**b**) Race tube assay showing the conidiation profile of WT and Δ*rme1* strains after 10 days of growth in MEX medium. (**c**) Growth profile of WT and Δ*rme1* strains in solid MAM containing different carbon sources. The same amounts of conidia (10^6^/mL) were inoculated in the center of the plates and incubated at 30 °C. Pictures were registered at the indicated time.

**Figure 3 jof-11-00658-f003:**
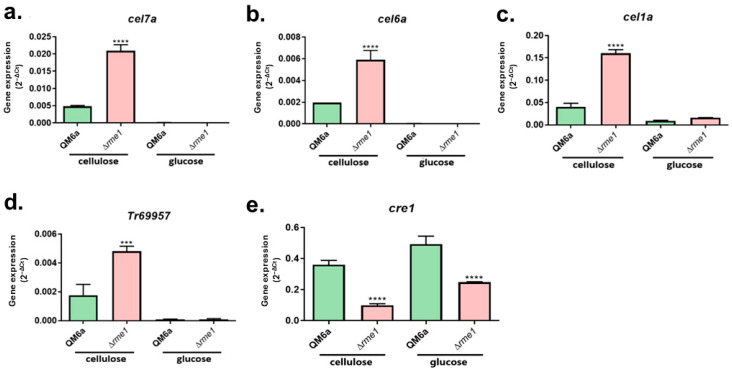
Rme1 regulates the expression of genes involved in biomass degradation. Gene expression analysis of the cellulolytic genes *cel7a* (**a**), *cel6a* (**b**), and *cel1a* (**c**), the sugar transporter *Tr69957* (**d**), and the transcription factor *cre1* (**e**), assessed by RT-qPCR. Gene expression was calculated using the 2^−ΔCt^ method using actin as the endogenous control. Strains were grown in MAM with cellulose for 24 h (after pregrowing in glycerol for 24 h) or MAM with glucose for 24 h. Asterisks indicate significant differences (*** *p* ≤ 0.001, **** *p* ≤ 0.0001) as assessed by One-way ANOVA followed by Bonferroni’s test.

**Figure 4 jof-11-00658-f004:**
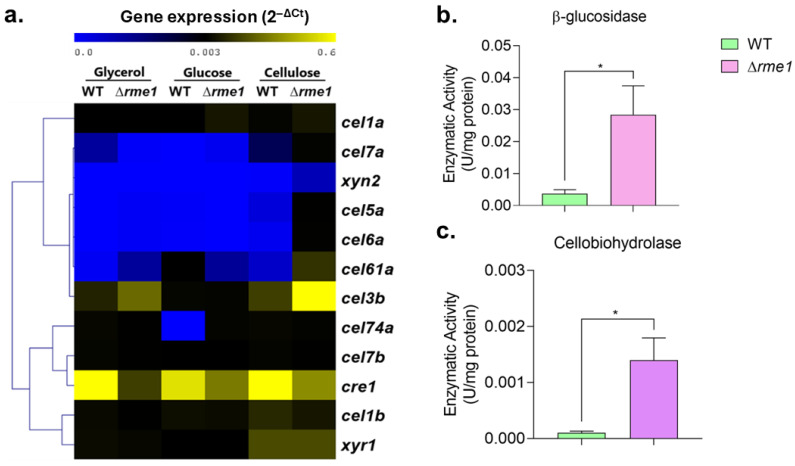
Rme1 is a repressor of holocellulase expression. (**a**) Heatmap showing the gene expression profile of holocellulolytic genes and the transcription factors *xyr1* and *cre1* in the WT and Δ*rme1* strains grown in different carbon sources. Gene expression was assessed using RT-qPCR. Gene expression was calculated using the 2^−ΔCt^ method using actin as the endogenous control. Heatmap was constructed using MEV, with the average linkage method for cluster generation. Strains were grown in MAM with cellulose (after pregrowing in glycerol for 24 h), glucose, or glycerol for 24 h. (**b**,**c**) Enzymatic activities of WT and Δ*rme1* strains after growth in cellulose. Proteins were extracted and used to measure β-glucosidase (**b**) and cellobiohydrolase (**c**) activities. Asterisks indicate significant differences (* *p* ≤ 0.05) as assessed by *t*-test followed by Welch’s test.

**Figure 5 jof-11-00658-f005:**
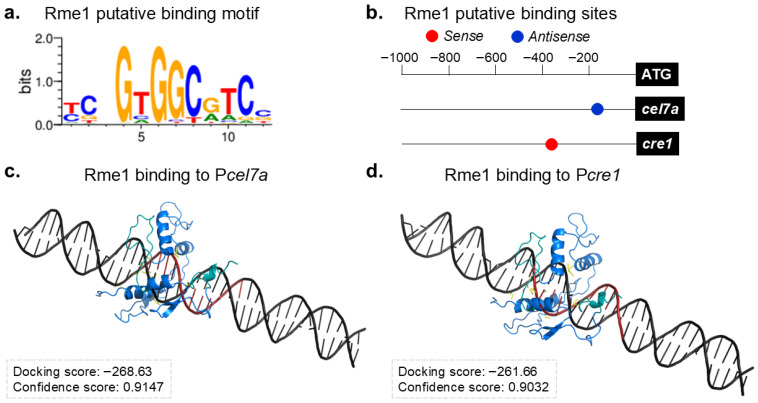
Identification of Rme1 direct target genes. (**a**) Rme1 putative binding motif. (**b**) Rme1 putative binding sites in the promoters of *cel7a* and *cre1* genes. (**c**,**d**) Molecular docking analyses of Rme1 interacting with the promoters of *cel7a* (**c**) and *cre1* (**d**) genes. The Rme1 zinc fingers are shown in cyan, and its binding motif in the promoters is highlighted in red.

**Figure 6 jof-11-00658-f006:**
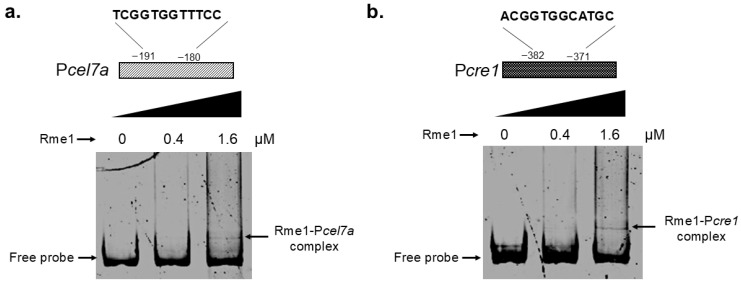
Rme1 binds to the promoters of *cel7a* and *cre1* genes. EMSA showing the interaction between the recombinant Rme1 and the P*cel7a* (**a**) and P*cre1* (**b**) promoters. Rme1:GST was expressed in *E. coli*, purified, and used to analyze the interaction with the promoters. Probes covering the Rme1 binding sites-containing regions of the promoters were used in the assay. Probe size is 300 bp.

**Figure 7 jof-11-00658-f007:**
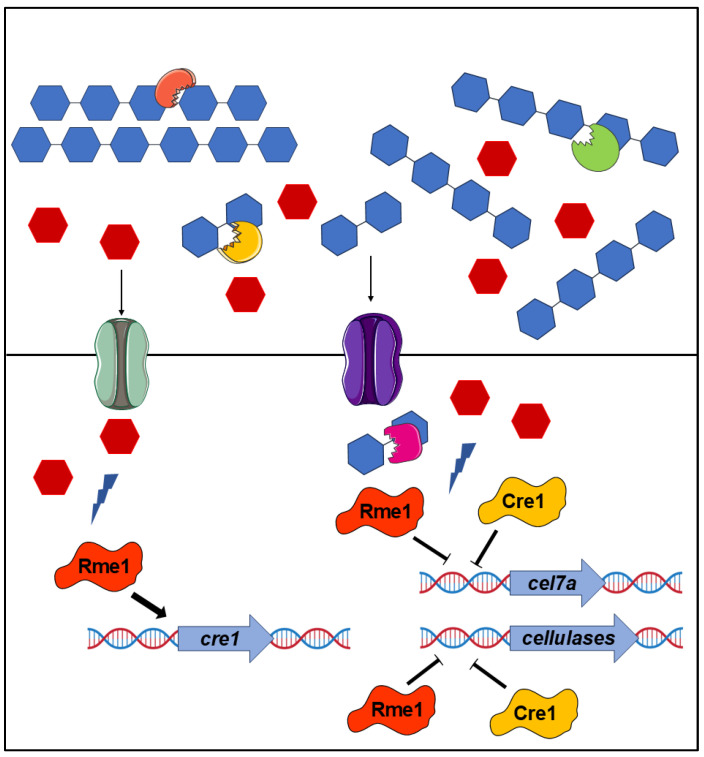
Model for Rme1-mediated regulation of holocellulase expression in *T. reesei*. The deconstruction of cellulose by the combined action of cellobiohydrolases, endoglucanases, and β-glucosidases releases glucose that causes the carbon catabolite repression. This glucose is imported into the cell and activates signaling pathways that activate Rme1. In turn, Rme1 activates the expression of *cre1* and, together, these two transcription factors repress the expression of *cel7a* and other hydrolytic genes.

## Data Availability

All relevant data are within this paper and its Appendix A files. Strains and plasmids are available upon request.

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
