# Peer review of "Rme1: Unveiling a Novel Repressor in the Cellulolytic Pathway of *Trichoderma reesei"

_jof, 2025, doi:10.3390/jof11090658_

Round 1

Reviewer 1 Report

This manuscript identifies a transcription factor, Rme1, in Trichoderma reesei that regulates the cellobiohydrolase Cel7a and the transcription factor Cre1. It also preliminarily investigates the physiological functions of this transcription factor both in vivo and in vitro. The study contributes to a better understanding of how Trichoderma reesei adapts to its environment and the mechanisms regulating cellulases expression. However, the reviewer believes the manuscript requires revision and improvement to more effectively support the study's conclusions.

1. The complemented strain needs to be constructed based on the Δrme1 strain. Phenotypic differences between the control strain, the knockout strain (Δrme1), and the complemented strain should be compared to further confirm the physiological functions of Rme1.

2. What is the effect of Rme1 on the growth of Trichoderma reesei on cellulose-containing medium? How does Rme1 affect the activities of various cellulases produced by Trichoderma reesei?

3. Statistical significance analysis is required for the expression levels shown in Figure 1d.

4. The EMSA results show weak binding of Rme1 to the probe. Was the probe used in the experiment only 10 bp? The authors should clearly state the probe length and other relevant information. Furthermore, the reviewer suggests adding additional in vivo experiments to verify whether Rme1 binds to the target sequence. If in vivo validation experiments proves difficult, at least additional in vitro experiments should be added to corroborate the EMSA results.

5. The overall quality of the English writing in the manuscript is poor. It is recommended to thoroughly proofread and polish the English language.

Author Response

Reviewer 1

Major comments

This manuscript identifies a transcription factor, Rme1, in Trichoderma reesei that regulates the cellobiohydrolase Cel7a and the transcription factor Cre1. It also preliminarily investigates the physiological functions of this transcription factor both in vivo and in vitro. The study contributes to a better understanding of how Trichoderma reesei adapts to its environment and the mechanisms regulating cellulases expression. However, the reviewer believes the manuscript requires revision and improvement to more effectively support the study's conclusions.

R: We thank the reviewer for the summary and the comments that certainly will improve the quality of our manuscript.

Detailed comments

  1. The complemented strain needs to be constructed based on the Δrme1 strain. Phenotypic differences between the control strain, the knockout strain (Δrme1), and the complemented strain should be compared to further confirm the physiological functions of Rme1.

R: We thank the reviewer for this comment. We agree that having a complemented strain will reinforce the role of Rme1 in T. reesei physiology. However, at the beginning of the characterization of the Δrme1 strain, we used three different transformants in some assays and all of them had a pretty similar phenotype. Therefore, if the reviewer doesn’t mind, we would like to not construct this strain. A complemented strain is under construction for further experiments of overexpression of this TF

  1. What is the effect of Rme1 on the growth of Trichoderma reesei on cellulose-containing medium? How does Rme1 affect the activities of various cellulases produced by Trichoderma reesei?

R: We thank the reviewer for this comment. We didn’t perform the growth on cellulose because it’s very difficult to do this assay on solid media as cellulose is insoluble and the results may not be reliable. However, in liquid medium no significant difference was observed between parental and mutant strains. However, all expression analyses were performed using transferred mycelium to exclude any interference of mutation in growth. We extracted the proteins from the mycelia after growing the strains on cellulose (liquid culture), and there is no difference between the WT and the Δrme1 strains (Figure R1).

We had some issues in measuring cellulase activity at the beginning but now we have added this assay to the manuscript. As shown in the new Figure 4, deletion of rme1 causes an increase in the β-glucosidase and cellobiohydrolase activities, in agreement with the gene expression results.

Figure R1. Protein concentration of the WT and Δrme1 strains. Strains were grown on cellulose for 48 h (after pregrowning on glycerol for 24 h) and the mycelia were collected for protein extraction.

  1. Statistical significance analysis is required for the expression levels shown in Figure 1d.

R: We thank the reviewer for this comment. We ran a statistical analysis for the gene expression shown in Fig 1d (One-way ANOVA followed by Bonferroni’s test). However, there are no statistical differences in the rme1 expression across the tested conditions, except for the difference between 8 and 24 h on cellulose. We rewrote this sentence in the text (page 8, lines 258-262): “There is no statistical difference in the expression levels between the majority of the tested conditions, however, rme1 expression significantly increases at 24 h of growth on cellulose, compared to the expression at 8 h on the same carbon source (p-value < 0.05, as accessed by One-way ANOVA followed by Bonferroni’s test).” We also include this in the new Figure 1d.

  1. The EMSA results show weak binding of Rme1 to the probe. Was the probe used in the experiment only 10 bp? The authors should clearly state the probe length and other relevant information. Furthermore, the reviewer suggests adding additional in vivo experiments to verify whether Rme1 binds to the target sequence. If in vivo validation experiments proves difficult, at least additional in vitro experiments should be added to corroborate the EMSA results.

R: We thank the reviewer for this comment. The probes used in the EMSA have 300 bp. We have added this information in the Material and Methods section and the figure caption. We acknowledge that the binding of Rme1 to DNA seems very weak in our assay, and we added this to the discussion (page 12, lines 453-460): “DNA-protein complexes were observed for both promoters, however, they needed a high amount of Rme1 to reduce their mobility, suggesting that Rme1 possibly has a low affinity for these promoters in vitro. Other experiments, such as Surface Plasmon Resonance and Chromatin Immunoprecipitation, can analyze the affinity of Rme1 to these promoters and their interaction in vivo, respectively. Despite this, the EMSA combined with the gene ex-pression and docking analyses, reinforces the notion that Rme1 can bind to the promoters of these genes to directly regulate their expression” We agree with the reviewer that in vivo experiments, such as ChIP-qPCR, would be great for proving the Rme1-DNA interaction and also the conditions under which it occurs. However, such experiments will performed further in overexpression mutants. This also applies to other in vitro experiments, such as SPR. Despite this, our EMSA, gene expression and docking analysis strongly suggest that Rme1 can bind to the promoters of cel7a and cre1, and regulates their expression. So, if the reviewer doesn’t mind, we would like to not do these experiments. Here, our interested was to validated the in silico analysis.   

  1. The overall quality of the English writing in the manuscript is poor. It is recommended to thoroughly proofread and polish the English language.

The manuscript was revised properly.

Reviewer 2 Report

In this manuscript, the authors identified a new transcription factor Rme1 which was involved in conidiation and cellulolytic gene expression of Trichoderma reesei. The overall methodology makes sense to me, but some data presenting details should be optimized.

  • Fig 1D, the bar of glucose 8h is missing, why?
  • From this manuscript, the function of Rme1 lies in two ways: regulate conidiation and cellulolytic gene expression. Are the two aspects related, or possibly related? This should be discussed.
  • Fig 3 and Fig 4. Both are RT-qPCR data, then why make two figures? Some genes in Fig 3 are also included in Fig 4, such as cel6a, cel1a and cre1, but looks not consistent. For example, cel1a under cellulose has a 3 folds elevation in Δrme1 strain (Fig 3), but disappeared in Fig 4. Please explain this.
  • Fig 4, what is the unit of the value 0-0.6? The legend mentioned MEV, but never explained.
  • Fig 6, EMSA assay. How to make sure that the tiny weak band is Rme1-probe complex? Any controls by specific and unspecific competitor?
  • Based on the discovery focusing on Rme1 in this manuscript, the authors should at least propose some strategies to engineer and help elevate cellulolytic enzyme expression.

See above

Author Response

Reviewer 2

Major comments

In this manuscript, the authors identified a new transcription factor Rme1 which was involved in conidiation and cellulolytic gene expression of Trichoderma reesei. The overall methodology makes sense to me, but some data presenting details should be optimized.

R: We thank the reviewer for the summary and the comments that certainly will improve the quality of our manuscript.

Fig 1D, the bar of glucose 8h is missing, why?

R: We thank the reviewer for this observation. We don’t analyze the gene expression at 8 h on glucose because we use conidia as inoculum and, after 8 h of culture, we don’t have enough mycelia to extract the RNA. While on cellulose or sugarcane bagasse, we use mycelia as inoculum, after pre-growing the fungus on glycerol (for glycerol, we also use conidia as inoculum). We rewrote the figure caption to make this information clear.

From this manuscript, the function of Rme1 lies in two ways: regulate conidiation and cellulolytic gene expression. Are the two aspects related, or possibly related? This should be discussed.

R: We thank the reviewer for this observation. Although there are some TFs that regulate conidiation and cellulase gene expression in T. reesei, the relationship between these two processes (if it exists) hasn’t been elucidated yet. We added this to the discussion (page 11, lines 422-428): “In T. reesei, several factors affect conidiation, such as light, pH, calcium, and carbon, nitrogen, and oxygen availability [29]. Some transcription factors have a role in regulating both conidiation and holocellulase expression in T. reesei, such as Vel1 [30] and Azf1 [11], besides Rme1 itself. Although the relationship between these two physiological processes needs to be deeply investigated, these results suggest that these processes can be controlled by the same regulators, probably as a response to nutrient status in the environment.”

Fig 3 and Fig 4. Both are RT-qPCR data, then why make two figures? Some genes in Fig 3 are also included in Fig 4, such as cel6a, cel1a and cre1, but looks not consistent. For example, cel1a under cellulose has a 3 folds elevation in Δrme1 strain (Fig 3), but disappeared in Fig 4. Please explain this.

R: We thank the reviewer for this comment. We split these results into two figures because they are from two different replicates. In the second (Figure 4), we included another carbon source (glycerol) and analyzed more genes. Although the gene expression values for some genes are not consistent with the first replicate, the biological conclusion remains (e.g. cel1a expression increases in Δrme1 strain). In addition, we changed the Figure 4 and included the enzymatic activities results.

Fig 4, what is the unit of the value 0-0.6? The legend mentioned MEV, but never explained.

R: We thank the reviewer for this comment. We are sorry that we forgot to explain this in the manuscript. The values represent the gene expression, accessed by the 2-ΔCt method using actin as endogenous control. MEV was used to generate the heatmap. We have fixed this figure and added the relevant information to the figure caption.

Fig 6, EMSA assay. How to make sure that the tiny weak band is Rme1-probe complex? Any controls by specific and unspecific competitor?

R: We thank the reviewer for this comment. Unfortunately, we weren’t able to use a specific competitor because in our assay we don’t label the probe (we amplify the probes by PCR, run the interaction assay and stain the DNA with SYBR Gold). Furthermore, we only see the Rme1-DNA complex when we use a high amount of protein, so we didn’t have enough protein to run a control with an unspecific competitor. We added this to the Discussion section (page 12, lines 453-460): “DNA-protein complexes were observed for both promoters, however, they needed a high amount of Rme1 to reduce their mobility, suggesting that Rme1 possibly has a low affinity for these promoters in vitro. Other experiments, such as Surface Plasmon Resonance and Chromatin Immunoprecipitation, can analyze the affinity of Rme1 to these promoters and their interaction in vivo, respectively. Despite this, the EMSA combined with the gene expression and docking analyses, reinforces the notion that Rme1 can bind to the promoters of these genes to directly regulate their expression.”

Based on the discovery focusing on Rme1 in this manuscript, the authors should at least propose some strategies to engineer and help elevate cellulolytic enzyme expression.

R: We thank the reviewer for this comment. We followed the reviewer’s suggestion and added this to the discussion (page 13, lines 492-500): “Our work raised Rme1 as a novel target for T. reesei engineering to enhance cellulase production. Deletion of this TF elevates cellulase production, with a highlight for β-glucosidases, the least produced enzyme by T. reesei. The protein-DNA results can be used to drive strategies such as promoter engineering and transcription factor engineering by domain fusion. For example, replacement of the Rme1 binding site in the cel7a promoter or fusion of the Rme1 DNA-binding domain to a trans-activator domain can increase cel7a expression. This can lead to strains with higher cellulase production.”

Reviewer 3 Report

The manuscript identified new role for the transcrption factor, Rme1 that involved in regulating gene expression of cellulase and Cre1. The results were well present and convincing. I have only few minor comments (see the detailed comments)

There are two motifs in Fig1a. However, they look very different and are not similar as the description in the manuscript. Please clarify it.

The phylogenetic tree in Fig. 1b lacks bootstrap values. And it can include more homologs from more commonly studied fungal species.

Why there is no grow profile of T. ressei grown on cellulose?

I am not convinced about the role of glucose in activating Rme1, given that the Rme1 seems activate in conditions of T. reesei grown with glycerol and xylose. Please modify the Fig. 7 and text.  

Author Response

Reviewer 3

Major comments

The manuscript identified new role for the transcrption factor, Rme1 that involved in regulating gene expression of cellulase and Cre1. The results were well present and convincing. I have only few minor comments (see the detailed comments)

R: We thank the reviewer for the summary and the comments that certainly will improve the quality of our manuscript.

Detailed comments

There are two motifs in Fig1a. However, they look very different and are not similar as the description in the manuscript. Please clarify it.

R: We thank the reviewer for this comment. Our approach to identifying new regulators of cellulase expression was searching for motifs enriched in the promoters of holocellulolytic genes in our RNA-Seq datasets. We found the motif shown in Figure 1a. Next, we used TOMTOM (https://meme-suite.org/meme/tools/tomtom) to search for similar known motifs from Saccharomyces cerevisiae using the default parameters, which led to the Rme1p binding motif. Although the motifs are not highly conserved, it is possible to observe that both present the TCAAAA sequence. We have now stated this in text (page 3, lines 109-11): “Next, we used TOMTOM (https://meme-suite.org/meme/tools/tomtom) to search for simi-lar known motifs from S. cerevisiae using the default parameters [16], which led to the Rme1p binding motif.” And page 5, lines 235-237: “Although the found motif does not present a high degree of conservation with the Rme1p binding motif, it is possible to observe that both present the TCAAAA consensus sequence.” We used this approach to identify new potential regulators of cellulase production in T. reesei and we have been pretty successful so far.

The phylogenetic tree in Fig. 1b lacks bootstrap values. And it can include more homologs from more commonly studied fungal species.

R: We thank the reviewer for this comment. We have followed the reviewer's suggestion and reconstructed our phylogenetic tree with a few more fungal species and also added the bootstrap values.

Why there is no grow profile of T. ressei grown on cellulose?

R: We thank the reviewer for this comment. We didn’t perform the growth on cellulose because it’s very difficult to do this assay on solid media as cellulose is insoluble and the results may not be reliable. We extracted the proteins from the mycelia after growing the strains on cellulose (liquid culture), and there is no difference between the WT and the Δrme1 strains (Figure R1).

Figure R1. Protein concentration of the WT and Δrme1 strains. Strains were grown on cellulose for 48 h (after pregrowning on glycerol for 24 h) and the mycelia were collected for protein extraction.

I am not convinced about the role of glucose in activating Rme1, given that the Rme1 seems activate in conditions of T. reesei grown with glycerol and xylose. Please modify the Fig. 7 and text. 

R: We thank the reviewer for this comment and we are sorry for this misunderstanding. Indeed, Rme1 has a function when the fungus grows in glycerol, xylose and also in malt extract (conidiation). However, the model presented in Figure 7 and discussed in the text deals only with the role of Rme1 in regulating cellulase gene expression. We have stated this in the manuscript (page 12, lines 436-442): “Furthermore, our results showed that the function of Rme1 in regulating holocellulase expression is carbon source-dependent: this TF only represses the expression of these genes during biomass breakdown. In addition, the results indicate that the absence of Rme1 is not sufficient to activate the expression of cellulases without an inducer, as we didn’t observe an increase in the expression of cellulolytic genes in the Δrme1 strain when grown in glycerol or glucose.” We also rewrote the sentences discussing the model (page 13, lines 487-492): “To summarize the role of Rme1 in regulating cellulase expression, we propose a model where the glucose, released during the cellulose breakdown by the action of cellulases, activates a signaling pathway that, in turn, leads Rme1 to the promoter of cre1 to activate its expression, and, together, these two repressors bind to the promoters of holocellulases genes, such as cel7a, to repress their expression (Figure 7).” We apologize for not explaining our model properly and we hope the reviewer agrees to keep it this way in the manuscript.

Reviewer 4 Report

The research manuscript by Antoniêto et al., “Rme1: Unveiling a Novel Repressor in the Cellulolytic Pathway 2 of Trichoderma reesei,” explores a relevant area of study. However, the presented data, analysis, and resulting conclusions exhibit several shortcomings that require substantial revision.

Please provide a deeper analysis of the presence of Rme1 binding site in promoter regions and expressions (RNA-seq) as its contra logic that the same binding site is related to simultaneously increasing and decreasing expression of genes (ln 318 to 320) and only one of the cellulolytic genes. Does it have to do with sense or antisense of the binding motif in the promoter? How many genes do have the putative binding site in their promoters? How many with the cre1 motif? Several cellulolytic genes have the same expression profile in the mutant. Why should the influence on the expression of cel7a not only from the reduced expression of Cre1? All these questions could be easily cleared if a high throughput screening was used such as e.g. trans‐activation domain sequencing.

There should be at least negative controls and probably a positive synthetic one (putative motif) in the data of Fig.6.

In conclusion, I recommend the rejection of the manuscript until the mentioned issues are properly addressed.

Ln 111 “Protein sequences of T. reesei and other fungi were obtained from FungiDB” but no entry was found for Trire_27649. Did the authors mean “TRIREDRAFT_27649”?? The last is an identifiable gene/protein code in public databases… the first is not. Please use the correct gene/protein codes throughout the document.

Fig 3. No units in graphs? If cel7a is the most important cellulolytic gene how come its relative expression is 10x lower than cel1a?

There are no legends for supplementary figures and no indication of the number of the figures.

Author Response

Reviewer 4

Major comments

The research manuscript by Antoniêto et al., “Rme1: Unveiling a Novel Repressor in the Cellulolytic Pathway 2 of Trichoderma reesei,” explores a relevant area of study. However, the presented data, analysis, and resulting conclusions exhibit several shortcomings that require substantial revision.

R: We thank the reviewer for the summary and the comments that certainly will improve the quality of our manuscript.

Please provide a deeper analysis of the presence of Rme1 binding site in promoter regions and expressions (RNA-seq) as its contra logic that the same binding site is related to simultaneously increasing and decreasing expression of genes (ln 318 to 320) and only one of the cellulolytic genes. Does it have to do with sense or antisense of the binding motif in the promoter? How many genes do have the putative binding site in their promoters? How many with the cre1 motif? Several cellulolytic genes have the same expression profile in the mutant. Why should the influence on the expression of cel7a not only from the reduced expression of Cre1? All these questions could be easily cleared if a high throughput screening was used such as e.g. trans‐activation domain sequencing.

R: We thank the reviewer for this comment. We also found it very interesting that Rme1 has this dual role as activator and repressor. We don’t believe this is due to sense or antisense binding site, but probably due to protein-protein interactions and post-translational modifications. Unfortunately, we don’t have data to support this hypothesis. We performed bioinformatic analysis to search for Rme1 binding sites in the promoters of the genes analyzed in the gene expression analysis (gene list is in Figure 4A) and we only found cre1 and cel7a as putative Rme1 target genes. Unfortunately, we are not able to search for Cre1 binding sites in the promoters of the analyzed genes and this information is not in the literature. RNA-seq analysis showed that deletion of cre1 causes up-regulation of several cellulolytic genes (10.1016/j.fgb.2014.10.009). In this context, if the increase in cel7a expression was due to reduced cre1 expression and not because deletion of rme1, we believe that more cellulolytic genes will be up-regulated in the Δrme1 strain. In addition, our results indicate cel7a as a direct target of Rme1. Cre1 ChIP would be a great tool to analyze the occupancy of this TF on cel7a promoter in the Δrme1 strain but this experiment would be very time-consuming and we are not able to do this at the moment and it is not in the scope of this manuscript. Unfortunately, due to the time we have for revision of this manuscript and the costs, we are not able to do high-throughput analysis (e.g. RNA-Seq and ChIP-seq) to see how Rme1 target genes behave. We added this topic to the discussion (page 12, lines 460-471): “It is noteworthy that Rme1 can function as an activator or a repressor, depending on its target gene. This is probably due to specific signaling pathways and protein-protein interactions. The transcription factor Cre1/CreA/Mig1p is known for being a transcriptional repressor. In Aspergillus nidulans, an integrated RNA-Seq and ChIP-Seq analysis identified CreA target genes that are positively regulated by it [33], however, the mechanism by which CreA can activate their expression still needs to be investigated. Interestingly, the Cyc8/Tup1 transcriptional co-repressors, which are partners of CreA/Mig1p in the repression of its target genes, also interact with Cre1 in T. reesei and with Xyr1, the main activator of cellulase gene expression [34–36]. Furthermore, Cyc8/Tup1 are required for appropriate cellulase expression, functioning also as co-activators for Xyr1 [35]. These studies indicate that transcription factors and other DNA-binding proteins can have different regulatory functions depending on cellular context.”

There should be at least negative controls and probably a positive synthetic one (putative motif) in the data of Fig.6.

R: We thank the reviewer for this comment. Unfortunately, we weren’t able to use a specific competitor because in our assay we don’t label the probe (we amplify the probes by PCR, run the interaction assay and stain the DNA with SYBR Gold). Furthermore, we only see the Rme1-DNA complex when we use a high amount of protein, so we didn’t have enough protein to run a control with an unspecific competitor. We added this to the Discussion section (page 12, lines 453-460): “DNA-protein complexes were observed for both promoters, however, they needed a high amount of Rme1 to reduce their mobility, suggesting that Rme1 possibly has a low affinity for these promoters in vitro. Other experiments, such as Surface Plasmon Resonance and Chromatin Immunoprecipitation, can analyze the affinity of Rme1 to these promoters and their interaction in vivo, respectively. Despite this, the EMSA combined with the gene expression and docking analyses, reinforces the notion that Rme1 can bind to the promoters of these genes to directly regulate their expression.”

In conclusion, I recommend the rejection of the manuscript until the mentioned issues are properly addressed.

R: Now we have made all the corrections pointed out by the reviewers and discussed all the topics raised by them. We hope now our manuscript can be accepted.

Detailed comments

Ln 111 “Protein sequences of T. reesei and other fungi were obtained from FungiDB” but no entry was found for Trire_27649. Did the authors mean “TRIREDRAFT_27649”?? The last is an identifiable gene/protein code in public databases… the first is not. Please use the correct gene/protein codes throughout the document.

R: We thank the reviewer for noticing this. Indeed, TRIREDRAFT_27649 is the correct form to refer to this gene. We have fixed this throughout the manuscript.

Fig 3. No units in graphs? If cel7a is the most important cellulolytic gene how come its relative expression is 10x lower than cel1a?

R: We thank the reviewer for this observation. We have fixed Figures 3 and 4 to add the gene expression in the graphs and also corrected the figure captions to explain how the analyses were performed. Regarding the expression of cel7a, we believe this is a feature of the QM6aΔtmus53Δpyr4 strain, which was used as the genetic background because it’s easier to do genetic manipulations in this strain. The information that Cel7a is the most important cellulase produced by T. reesei came from proteomic analysis of the fungus secretome grown in different plant biomass (as reviewed here https://doi.org/10.1186/s13068-021-01955-5). We have been using a different T. reesei strain, with enhanced cellulolytic activity, for functional genomics in our group and cel7a is 10x more expressed than cel1a (Figure R2)

Figure R2. Gene expression analysis of cel7a and cel1a was assessed by RT-qPCR. T. reesei strain QM9414 was grown for 24 h in glycerol and, then, the mycelia were transferred to cellulose and cultivated for 24 h before being collected for RNA extraction.

There are no legends for supplementary figures and no indication of the number of the figures.

R: We thank the reviewer for noticing this. We have uploaded the wrong supplementary material. Now we have fixed it.

Round 2

Reviewer 1 Report

The author answered all the questions raised by the reviewer in the reply. In response to review comments 1 and 2, although the author provided further explanations and clarifications on the current experimental evidence, the author did not supplement the experiments proposed by the reviewers and did not present further experimental evidence. In response to other reviewer's comments, the author made necessary explanations and revisions.

No detailed comment.

Author Response

Reviewer 1

The author answered all the questions raised by the reviewer in the reply. In response to review comments 1 and 2, although the author provided further explanations and clarifications on the current experimental evidence, the author did not supplement the experiments proposed by the reviewers and did not present further experimental evidence. In response to other reviewer's comments, the author made necessary explanations and revisions.

R: Dear Reviewer, we thank for your patience in review our manuscript and your comments. We followed a suggestion for another reviewer and we ran an EMSA using a probe for the swo gene (not a target for Rme1). We chose this promoter because we already have primers to amplify it and this gene encodes a protein that acts in biomass degradation, which would better than the actin promoter. As shown in Figure S3 and R2.1, Rme1 doesn’t bind to the swo promoter. This result indicates that, although it seems weak, Rme1 interaction with Pcel7a and Pcre1 promoters is specific. We stated this now in the manuscript (page 10, lines 383-387): “As a negative control, the EMSA was performed using a probe for the swo gene and increasing concentrations of purified Rme1, however, no shift in mobility was observed (Figure S3), indicating no formation of protein-DNA complex  and that the interaction between Rme1 and Pcel7a and Pcre1 is specific.”

Figure R2.1. Rme1 doesn’t bind to the promoter of swo gene. EMSA was performed with increasing concentrations of purified Rme1:GST and a PCR-amplified probe for Pswo (140 bp).

Reviewer 2 Report

The authors have solved most of my concerns. The only problem still lies in Figure 6. If not enough proteins, then purify more. Probes could also be labeled by biotin easily. Moreover, other experiments such as ChIP-seq could also be used to check the binding between TF and promoters in vivo.

See above.

Author Response

Reviewer 2

The authors have solved most of my concerns. The only problem still lies in Figure 6. If not enough proteins, then purify more. Probes could also be labeled by biotin easily. Moreover, other experiments such as ChIP-seq could also be used to check the binding between TF and promoters in vivo.

R: Dear Reviewer, we thank for your patience in review our manuscript and your comments. We have purified more Rme1 and we performed an EMSA using a probe for the swo gene (not a target for Rme1). We chose this promoter because we already have primers to amplify it and this gene encodes a protein that acts in biomass degradation, which would better than the actin promoter. As shown in Figure S3 and R2.1, Rme1 doesn’t bind to the swo promoter. This result indicates that, although it seems weak, Rme1 interaction with Pcel7a and Pcre1 promoters is specific. We stated this now in the manuscript (page 10, lines 383-387): “As a negative control, the EMSA was performed using a probe for the swo gene and increasing concentrations of purified Rme1, however, no shift in mobility was observed (Figure S3), indicating no formation of protein-DNA complex and that the interaction between Rme1 and Pcel7a and Pcre1 is specific.”

Figure R2.1. Rme1 doesn’t bind to the promoter of swo gene. EMSA was performed with increasing concentrations of purified Rme1:GST and a PCR-amplified probe for Pswo (140 bp).

Now we are more interested in Rme1 and we are planning to conduct ChIP-Seq and RNA-seq to see how it targets behave but this is for another manuscript.

Reviewer 4 Report

I believe the authors have adequately addressed most of the reviewers' questions and have significantly improved the manuscript. However, I must insist that the EMSA experiment (Fig 6) needs negative control for instance with the PCR amplified promoter of other cellulolytic gene like cel1a. This is the most important result of the manuscript and at least a negative control should be presented. In conclusion, I still recommend the rejection of the manuscript until the mentioned issues are properly addressed.

Nothing to add.

Author Response

Reviewer 4

I believe the authors have adequately addressed most of the reviewers' questions and have significantly improved the manuscript. However, I must insist that the EMSA experiment (Fig 6) needs negative control for instance with the PCR amplified promoter of other cellulolytic gene like cel1a. This is the most important result of the manuscript and at least a negative control should be presented. In conclusion, I still recommend the rejection of the manuscript until the mentioned issues are properly addressed.

R: R: Dear Reviewer, we thank for your patience in review our manuscript and your comments. We followed the reviewer’s suggestion and we ran an EMSA using a probe for the swo gene (not a target for Rme1). We chose this promoter because we already have primers to amplify it and this gene encodes a protein that acts in biomass degradation, which would better than the actin promoter. As shown in Figure S3 and R2.1, Rme1 doesn’t bind to the swo promoter. This result indicates that, although it seems weak, Rme1 interaction with Pcel7a and Pcre1 promoters is specific. We stated this now in the manuscript (page 10, lines 383-387): “As a negative control, the EMSA was performed using a probe for the swo gene and increasing concentrations of purified Rme1, however, no shift in mobility was observed (Figure S3), indicating no formation of protein-DNA complex. and the interaction between Rme1 and Pcel7a and Pcre1 is specific.”

Figure R2.1. Rme1 doesn’t bind to the promoter of swo gene. EMSA was performed with increasing concentrations of purified Rme1:GST and a PCR-amplified probe for Pswo (140 bp).

Round 3

Reviewer 2 Report

My problem is solved. No more questions.

None.

Reviewer 4 Report

The authors have adequately answered my last recommendation to improve the results presentation with the addition of a relevant negative control, and I therefore recommend this manuscript for publication.

Nothing to add.